# GENERATIVE PRE-TRAINING FOR SPEECH WITH FLOW MATCHING

**Alexander H. Liu**[1],[*] **Matt Le**[2]**, Apoorv Vyas**[2]**, Bowen Shi**[2]**, Andros Tjandra**[2]**, Wei-Ning Hsu**[2]
[1]MIT CSAIL, [2]Meta AI
[1]alexhliu@mit.edu

## ABSTRACT

Generative models have gained more and more attention in recent years for their remarkable success in tasks that required estimating and sampling data distribution to generate high-fidelity synthetic data. In speech, text-to-speech synthesis and neural vocoder are good examples where generative models have shined. While generative models have been applied to different applications in speech, there exists no general-purpose generative model that models speech directly. In this work, we take a step toward this direction by showing a single pre-trained generative model can be adapted to different downstream tasks with strong performance. Specifically, we pre-trained a generative model, named SpeechFlow, on 60k hours of untranscribed speech with Flow Matching and masked conditions. Experiment results show the pre-trained generative model can be fine-tuned with task-specific data to match or surpass existing expert models on speech enhancement, separation, and synthesis. Our work suggested a foundational model for generation tasks in speech can be built with generative pre-training. Audio samples can be found at https://voicebox.metademolab.com/speechflow.html.

## 1 INTRODUCTION

Discriminative models have long been the mainstream in speech applications since the deep learning era. These models are applied to different types of tasks such as speech recognition (Graves et al., 2006), enhancement, and separation (Luo & Mesgarani, 2019). Interestingly, even for applications that can be naturally formulated as generative modeling problems, such as text-to-speech (TTS), we see most popular models remained discriminative (Shen et al., 2018; Ren et al., 2021). Consequentially, pre-trained foundation models (Baevski et al., 2020; Hsu et al., 2021) that served as the upstream of speech applications focused more on learning useful representation for discriminative tasks rather than modeling the data distribution $p(\text{speech})$. In this paper, we seek to answer *whether generative models can serve as foundation models for speech applications or not*.

Unlike discriminative models, generative models enable sampling of the data distribution. For example, generative TTS models (Habib et al., 2019) allow different emotions to be sampled given a fixed text as discriminative models produce a fixed output. Up to the present, generative models in speech are usually designed for a given purpose via *task-specific conditioning* or *distribution mapping*. Perhaps the most well-known examples of task-specific conditional generative models are neural vocoders (Kong et al., 2020; Chen et al., 2020). These models learn to map simple priors (e.g., normal distribution) to waveform conditioning on acoustic features (e.g., spectrogram). On the other hand, examples for distribution mapping include diffusion models that transform noisy speech to clean speech for denoising (Lu et al., 2021; 2022; Richter et al., 2023), or speech mixture to non-overlapping speech for separation (Scheibler et al., 2023).

In this work, we explore a new direction to pre-train a general-purpose generative model with unlabeled speech. We hypothesize that a good generative model on speech without pre-defined application can be applied to different end tasks that require speech generation. Our model, named SpeechFlow, is a generative model that combines masked audio modeling and Flow Matching (Lipman et al., 2023). SpeechFlow is trained with unlabeled speech with the goal of estimating the

---

*Work done during an internship at Meta.

underlying distribution of speech conditioning on masked audio. We show that a generative model trained with unlabeled speech data can be adapted to different tasks that require speech generation by fine-tuning with task-specific conditions using labeled data. More specifically, we fine-tuned SpeechFlow and compared against expert models in speech enhancement, separation, and synthesis. For each task, fine-tuned SpeechFlow is able to match expert models. Experiment results suggested that pre-trained generative models possess great potential to become foundation models for different speech generation tasks.

## 2 RELATED WORK

**Generative Speech Models**  As mentioned earlier, generative models have been applied to different tasks in speech. Research in neural vocoders found generative models to be a good suit for spectrogram-to-waveform prediction. Prevailing generative models are applied to the task with success, such as generative adversarial model (Kong et al., 2020), flow-based invertible model (Prenger et al., 2019), and diffusion network (Koizumi et al., 2022). Besides neural vocoders, generative models are also applied to other tasks such as TTS (Valle et al., 2020), speech enhancement (Lu et al., 2021; 2022; Richter et al., 2023) and separation Scheibler et al. (2023). A fundamental difference between this work and the prior works is that SpeechFlow is *not* trained for a specific application, but to estimate the underlying distribution of speech itself.

Recent studies also explored speech generation from a language modeling perspective. Taking advantage of audio tokenizing techniques (Hsu et al., 2021; Défossez et al., 2022; Zeghidour et al., 2022), Spoken Language Models (SLMs;Lakhotia et al., 2021; Kharitonov et al., 2021; Borsos et al., 2022) have been developed to model language without text. These token-based speech language models are closely related to the proposed method in the sense of training generative models from unlabeled speech. The key difference is the goal of SLMs is to discover the underlying text for textless language processing (Nguyen et al., 2022). In principle, SLMs can also be fine-tuned for different downstream tasks but it was not the focus and they are not evaluated on multiple tasks.

Targeting controllable audio generation, VALL-E (Wang et al., 2023) extended SLMs by using text and audio prompts to control the audio generated. Voicebox (Le et al., 2023) took a different approach to tackle the problem by feeding aligned text and partially masked speech to perform speech in-filling non-autoregressively. Despite the different paths VALL-E and Voicebox took, both works discovered a strong zero-shot adaptation ability that emerged when training generative models at scale. While these models are designed for text-conditioned generation, they provided a hint of the great potential of generative models with the superior ability to generate diverse speech. It is worth pointing out that Voicebox is the most related work to this work, sharing the same objective function and model architecture. Voicebox can be viewed as a fully supervised text-conditioned SpeechFlow that focused exclusively on TTS task. Later in our experiment, we compare Voicebox to fine-tuned SpeechFlow and reveal the benefit of generative pre-training without text.

**Pre-trained Speech Models**  Conceptually, this work is also related to self-supervised representation learning methods for speech in the sense of learning from unlabeled data for better downstream task performance. One branch of self-supervised learning takes the autoregressive approach to learn from predicting the future, such as contrastive predictive coding (Oord et al., 2018) and autoregresive predictive coding (Chung & Glass, 2020). Another branch of works (Ling et al., 2020; Ling & Liu, 2020) studied masked audio modeling (MAM) instead of future prediction. These models predict masked Spectrogram based on the complementary part of the input that is unmasked. Improving the MAM-based method, similar works replaced the prediction target with latent features such as quantized representation (Baevski et al., 2020) or acoustic units (Hsu et al., 2021). Self-supervised representation learning methods are found to be useful in many different applications such as speech recognition (Yang et al., 2021). But the success is mostly on discriminative tasks, applying self-supervised models for generation application tasks is often less intuitive (Polyak et al., 2021) and under-performing (Tsai et al., 2022). Taking cues from the success of masking-based methods, we incorporate a similar idea into SpeechFlow  to make generation conditioned on partially masked speech during pre-training. Interestingly, we found MAM beneficial to generative pre-training as shown later in Section A.4.5. Besides self-supervised learning, pre-training have also been studied in the context of semi-supervised TTS (Chung et al., 2019) or speech-text alignment (Ao et al., 2021), but these works focused on non-generative models.

## 3 METHOD

### 3.1 BACKGROUND: FLOW MATCHING FOR GENERATIVE MODELING

Deep generative models aimed to estimate the unknown distribution $q(x)$ of real world $d$-dimensional data $x \in \mathbb{R}^d$ with distribution $p(x)$ parameterized by neural networks. To make sampling possible, simple prior distribution $p_0(x)$ (e.g., normal distribution) is naturally a good starting point, and the modeling problem therefore becomes finding a neural transport map $p_1 = F_\theta(p_0)$ such that $p_1(x) \approx q(x)$. Early works such as generative adversarial networks (Goodfellow et al., 2020) and variational audio encoders (Kingma & Welling, 2013) showed directly modeling $x_1 = f_\theta(x_0)$ where $x_0 \sim p_0(x), x_1 \sim q(x)$, i.e., predicting data from noise using network $f_\theta$, is feasible. Recent studies in diffusion models (Ho et al., 2020; Song et al., 2020) suggested an iterative denoising model $x_{t+\Delta t} = f_{\theta,t,\Delta t}(x_t)$ that traverses from noise $x_0$ to data $x_1$ with step size $\Delta t$ provides better generation quality (Dhariwal & Nichol, 2021). In this work, we choose to construct the neural transport map $p_1 = F_\theta(p_0)$ using Flow Matching (Lipman et al., 2023) from the Continuous Normalizing Flows (CNFs; Chen et al., 2018)- family.

Formally, CNFs defined a *path* between simple prior $p_0$ and target distribution $p_1$ via the time-dependent probability density function $p_t : [0, 1] \times \mathbb{R}^d \to \mathbb{R}_{>0}$. The *flow* of $x$ along the path, denoted $\phi_t : [0, 1] \times \mathbb{R}^d \to \mathbb{R}^d$, is defined using ordinary differential equation (ODE):

$$\frac{d}{dt}\phi_t(x) = v_t(\phi_t(x)); \quad \phi_0(x) = x; \tag{1}$$

with the time-dependent vector field $v_t : [0, 1] \times \mathbb{R}^d \to \mathbb{R}^d$, such that the time-dependent probability density function $p_t$ can be derived using the change of variables formula: $p_t = p_0(\phi_t^{-1}(x)) \det \left[ \frac{\partial \phi_t^{-1}}{\partial x}(x) \right]$. Under the formulation, a simple objective is to predict the vector field $v_t$ using a neural network paramterized by $\theta$ given the target vector field $u_t(x)$ that corresponds to $p_t(x)$ with the Flow Matching objective

$$\mathcal{L}_{FM}(\theta) = \mathbb{E}_{t\sim\mathcal{U}[0,1],x\sim p_t(x)}\left\| v_t(x;\theta) - u_t(x) \right\|^2. \tag{2}$$

However, $\mathcal{L}_{FM}(\theta)$ is intractable due to the lack of knowledge of $p_t$ and $u_t$ in practice. Interestingly, Lipman et al. (2023) showed that conditioning $p_t$ and $u_t$ on real data $x_1$ results in the Conditional Flow Matching objective $\mathcal{L}_{CFM}(\theta)$ which provided identical gradient w.r.t. $\theta$ for training the generative model. Specifically, we adopt the Optimal Transport conditional path proposed by Lipman et al. (2023) that assumes the mean $\mu_t(x) = tx_1$ and standard deviation $\sigma_t(x) = 1 - (1 - \sigma_{\min})t$ change linearly in time, yielding tractable $p_t(x|x_1) = \mathcal{N}(x \mid \mu_t(x_1), \sigma_t(x_1)^2 I)$ and $u_t(x|x_1) = \frac{(x_1 - (1-\sigma_{\min})x)}{(1-(1-\sigma_{\min})t)}$ with a sufficiently small $\sigma_{\min}$ (we use 1e-5) such that $p_1(x|x_1)$ is centered around $x_1$. In this case, with reparameterization the Conditional Flow Matching objective has the form

$$\mathcal{L}_{CFM}(\theta) = \mathbb{E}_{t,q(x_1),p_0(x_0)}\left\| v_t(\psi_t(x_0);\theta) - \left(x_1 - (1 - \sigma_{\min})x_0\right) \right\|^2, \tag{3}$$

where $\psi_t(x_0) = \sigma_t(x_1)x_0 + \mu_t(x_1)$ and $t$ is sampled uniformly from $[0, 1]$.

### 3.2 GENERATIVE PRE-TRAINING OF SPEECHFLOW WITH UNLABELED SPEECH

Inspired by the recent success of flow matching model in speech synthesis (Le et al., 2023), we propose to pre-train a generative model with unlabeled speech using flow matching. We consider the problem of modeling $q(x)$ where the acoustic features $x \in \mathbb{R}^{d \times L}$ are $d$-dimensional Mel spectrogram with $L$ frames. We assume the simple prior $p_0$ to be the normal distribution. Since generative models are by nature unsupervised/self-supervised (no human label required), a flow matching model can be trained with pure speech.

**Masked Audio Condition**  In light of the success of masked prediction in self-supervised speech representation learning (Baevski et al., 2020; Hsu et al., 2021), we introduce similar concept to SpeechFlow by additionally conditioning $v_t$ on partially masked target audio $x_{\text{mask}}$ with a chance of $p_{\text{cond}}$ during training. This can also be interpreted as the model have a chance of $1 - p_{\text{cond}}$ to receive

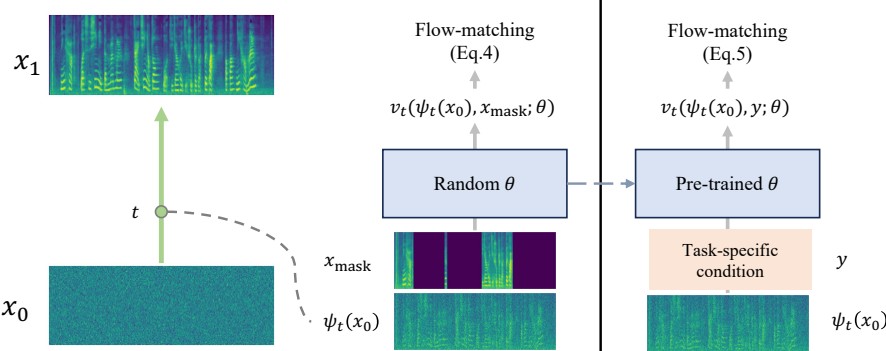

Figure 1: An overview of SpeechFlow. (Left) Pre-training with masked audio. (Right) Fine-tuning with task-specific condition such as noisy recording, overlapped speech, or phone sequence. More details of the model and conditioning are available in Section A.3.

fully masked $x_{\text{mask}}$. Masked condition $x_{\text{mask}}$ is obtained by randomly selecting $n_{\text{mask}}$ of frames to be masked with a minimum masking span length of $l_{\text{mask}}$.

Note that while this modification results in a conditional generative model, our model is still self-supervised since $x_{\text{mask}}$ is directly derived from unlabeled speech $x_1$. Moreover, a vanilla flow matching model without any condition is still available after pre-training stage as long as $p_{\text{cond}} < 1$. Study on the importance of $p_{\text{cond}}$ is provided in Section A.4.5.

The rationale behind the auxiliary condition is to provide the model more context for predicting $v_t$ regardless of the timestep $t$. Moreover, introducing auxiliary condition at the pre-training stage provided an intuitive way to fine-tune the model for different tasks as shown later in this section.

**Objective**   With the predicted time-dependent vector field $v_t$ conditioning on masked feature $x_{\text{mask}}$, the generative pre-training objective of SpeechFlow can be derived by modifying Equation 3 accordingly to obtain

$$\mathbb{E}_{t,q(x_1),p(x_0)}\left\|v_t(\psi_t(x_0), x_{\text{mask}}; \theta) - \left(x_1 - (1 - \sigma_{\min})x_0\right)\right\|^2. \tag{4}$$

In practice, we use Transformer encoder (Vaswani et al., 2017) with learnable parameter $\theta$ to predict vector field $v_t$. Masked inputs $x_{\text{mask}}$ are concatenated with $\psi_t(x_0)$ along the frequency axis, then projected to match the model dimension $d_\theta$, and we append the sinusoidal positional encoding of timestep $t$ to the input, resulting the actual model input with shape $\mathbb{R}^{d_\theta \times (L+1)}$. The output of the model is the predicted vector field $v_t \in \mathbb{R}^{d \times L}$.

### 3.3   Supervised Fine-tuning SpeechFlow on Different Tasks

**Task-specific Condition**   While the pre-trained SpeechFlow allow us to sample new data from $p_1(x)$, most applications in speech require a certain degree of control over the output. To this end, we introduce the fine-tuning stage for controllable generation using task-specific condition $y \in \mathbb{R}^{d_y \times L_y}$ of audio $x_1$, such as noisy speech for speech enhancement and text transcript for text-to-speech generation. We note that this work focused on tasks where $y$ and $x_1$ are aligned, i.e., $L_y = L$, and leave the unaligned cases for future work. Concrete examples can be found in Section A.3.

**Objective**   Following the pre-training stage, the fine-tuning objective can be derived by swapping the masked condition $x_{\text{mask}}$ for pre-training with task-specific condition $y$,

$$\mathbb{E}_{t,q(x_1),p(x_0)}\left\|v_t(\psi_t(x_0), y; \theta) - \left(x_1 - (1 - \sigma_{\min})x_0\right)\right\|^2. \tag{5}$$

Note that for fine-tuning, it is critical to reuse $\theta$ from the pre-training stage.

**Inference**   After training, speech generation is done by the following steps: (1) sample $x_0$ from the simple prior $p_0(x)$; (2) use an ODE solver to solve $\phi_1(x_0)$ given $d\phi_t(x_0)/dt = v_t(\phi_t(x_0), y; \theta)$ and $\phi_0(x_0) = x_0$; (3) generated audible speech in time domain from Mel spectrogram $x_1$. More inference details are provided in Section A.2 including conversion from Mel spectrogram to waveform.

Table 1: Speech enhancement test results on Voicebank-Demand (Valentini-Botinhao et al., 2017) and WSJ0-CHiME3 (Richter et al., 2023). Best result of each section is **bolded**. Numbers are taken from prior works unless otherwise specified. For full result that includes more metrics, please refer to Table 7.

| Method | Voicebank-Demand | | | | WSJ0-CHiME3 | | | |
|---|---|---|---|---|---|---|---|---|
| | PESQ | ESTOI | CSIG | COVL | PESQ | ESTOI | CSIG | COVL |
| Baseline | | | | | | | | |
|   Mixture | 1.97 | 0.79 | 3.35 | 2.63 | 1.69 | 0.78 | 3.24 | 2.42 |
| Models trained on Voicebank-Demand | | | | | | | | |
|   Conv-TasNet[†](Luo & Mesgarani, 2019) | 2.63 | 0.85 | - | - | 2.40 | 0.88 | - | - |
|   MetricGAN+ (Fu et al., 2021) | **3.13** | 0.83 | 4.10[*] | 3.61[*] | 2.13 | 0.76 | 3.02[*] | 2.52[*] |
|   SGMSE+ (Richter et al., 2023) | 2.93 | **0.87** | 4.13[*] | 3.53[*] | 2.48 | **0.90** | 3.67[*] | 3.02[*] |
|   SpeechFlow | **3.13** | **0.87** | **4.43** | **3.80** | **2.70** | **0.90** | **4.05** | **3.36** |
|   SpeechFlow w/o pre-train | 2.92 | 0.85 | 4.22 | 3.57 | 2.38 | 0.86 | 3.72 | 3.03 |
| Models trained on Deep Noise Supression Challange 2020 (Reddy et al., 2020) | | | | | | | | |
|   DEMUCS | 2.55[*] | 0.85[*] | 3.24[*] | 2.88[*] | 2.49[*] | **0.92**[*] | 3.93[*] | 3.20[*] |
|   SpeechFlow | **2.71** | **0.86** | **4.07** | **3.39** | **2.87** | 0.91 | **4.24** | **3.54** |
|   SpeechFlow w/o pre-train | 2.53 | 0.84 | 3.89 | 3.20 | 2.56 | 0.89 | 3.91 | 3.22 |
| Topline | | | | | | | | |
|   Our upper-bound[‡] | 3.77 | 0.95 | 4.97 | 4.54 | 3.68 | 0.96 | 4.97 | 4.46 |
|   Clean signal | 4.50 | 1.00 | 5.00 | 5.00 | 4.50 | 1.00 | 5.00 | 5.00 |

[*] Results reproduced by us using the open sourced model released by the authors.
[†] Results reproduced by Richter et al. (2023).
[‡] Clean Mel spectrogram with error introduced by pseudo-inversing Mel filter bank and taking phase from the mixture.

# 4 EXPERIMENT

## 4.1 PRE-TRAINING DETAILS

**Model & Data** We focus on Transformer encoder (Vaswani et al., 2017) with 24 layers, 16 attention heads, $d_\theta = 1024$ dimensional embedding, and feed-forward networks with 4096 dimensions. Convolutional positional embedding (Baevski et al., 2020) and ALiBi self-attention bias (Press et al., 2021) are used to encode relative positional information. Following Le et al. (2023), skip connections between layers are introduced to mimic U-Net (Ronneberger et al., 2015) architecture. The model has around 330M parameters in total. The model is pre-trained on 60k hours of speech from English audiobook at 16kHz. We consider $x$ to be log-scaled Mel spectrogram extracted with a 40ms window at 100Hz with $d = 80$, resulting $160/80$ dimensional input/output for the model.

**Training** We pre-train SpeechFlow for 600k steps on 32 V100 GPUs with a batch size of 75 seconds per GPU with FP16. We use Adam optimizer (Kingma & Ba, 2014) with the learning rate warming up linearly to 5e-5 for the first 5k steps and linearly decaying to 1e-5 for the rest of the training. For masking, we set $p_{\text{drop}} = 10\%$, $n_{\text{mask}} \sim \mathcal{U}[70\%, 100\%]$, and $l_{\text{mask}} = 10$. All masked position are filled with zero. In practice, we compute loss at the masked position only.

## 4.2 FINE-TUNING FOR SPEECH ENHANCEMENT

**Task & Metrics** Speech enhancement, also known as denoising, aimed to remove unwanted noise from speech recording. We report Perceptual Evaluation of Speech Quality (PESQ; Rix et al., 2001), Extended Short-Time Objective Intelligibility (ESTOI; Jensen & Taal, 2016), and Composite Objective Speech Quality and Overall Quality (CSIG/COVL;Hu & Loizou, 2007).

**Prior Works** Early work Conv-TasNet (Luo & Mesgarani, 2019) has been widely used as the baseline system. It is a convolutional encoder/decoder operating in the time domain to maximize scale-invariant source-to-noise ratio. DEMUCS (Défossez et al., 2020) adopted a similar structure with skip-connections and minimized L1/multi-resolution STFT loss. MetricGAN+ (Fu et al., 2021) proposed to optimize non-differentiable metrics such as PESQ via adversarial training against their approximation using discriminators. SGMSE+(Richter et al., 2023) reformulated the problem as a diffusion process that can be solved with the corresponding generative model (Ho et al., 2020).

**Dataset** We fine-tuned and tested SpeechFlow on the benchmark dataset VoiceBank-Demand (VB-DMD; Valentini-Botinhao et al., 2017) for fair comparison against most of the prior works

Table 2: Speech separation test results on LibriMix (Cosentino et al., 2020). All models are trained on 16kHz audio without data augmentation. Best model output for each metric is **bolded**.

| Method | 2 Mix | | 2 Mix + Noise | | 3 Mix | | 3 Mix + Noise | |
|---|---|---|---|---|---|---|---|---|
| | SI-SDRi | ESTOIi | SI-SDRi | ESTOIi | SI-SDRi | ESTOIi | SI-SDRi | ESTOIi |
| Conv-TasNet[†] | 15.24 | 0.22 | **12.55** | 0.22 | **12.30** | 0.26 | **10.28** | 0.21 |
| SepFormer[‡] | 14.94 | 0.31 | 11.71 | 0.28 | - | - | - | - |
| *Pseudo-inversed Mel and phase from mixture* | | | | | | | | |
| Upper-bound w/ clean Spec. | 12.43 | 0.35 | 11.99 | 0.46 | 12.91 | 0.44 | 12.62 | 0.48 |
| SpeechFlow | 11.74 | 0.35 | 10.46 | 0.33 | 11.08 | **0.35** | 8.22 | **0.23** |
| SpeechFlow w/o pre-train | 11.24 | 0.29 | 10.00 | 0.31 | 8.65 | 0.24 | 7.39 | 0.19 |
| *Learnable inverse-Mel and phase estimation* (See Section A.2 for more details.) | | | | | | | | |
| SpeechFlow | **15.85** | **0.37** | 12.41 | **0.37** | - | - | - | - |

[†] Luo & Mesgarani (2019), reproduced by Cosentino et al. (2020). [‡] Subakan et al. (2021; 2023), reproduced at 16kHz with official code from SpeechBrain (Ravanelli et al., 2021), note that this method was originally designed for 8kHz audio with data augmentation.

in the field. Since VB-DMD is a relatively small dataset, we also consider testing on WSJ0-CHiME3 (Richter et al., 2023) to ensure the model is not overfitting. In addition, we also trained our model using 100 hours of noisy speech from Deep Noise Supression Challenge 2020 (DNS2020; Reddy et al., 2020) for extra results to demonstrate the generalizability for SpeechFlow. For training, paired data $(x_1, y)$ is provided where $x_1$ is the target clean signal and $y$ is the noisy speech. For testing, only the noisy speech $y$ is provided and the goal is to estimate the clean signal $x_1$. All datasets are resampled to 16kHz to match pre-training and no data augmentation was applied.

**Training** As mentioned in Section 3.3, fine-tuning is simply done by replacing the auxiliary masked condition $x_m$ for pre-training with the acoustic feature of the noisy speech $y$ and minimize Eq. 5. Note that, unlike pre-training, $y$ has a $p_{\text{drop}} = 30\%$ chance to be dropped but never partially masked for fine-tuning. We fine-tuned SpeechFlow on single V100 GPU for 160 / 75 epochs on VB-DMD / DNS2020 respectively with a batch size of 50 seconds. The learning rate is set to peak at 2e-5 after 5k updates, then linearly decay to 0. For the control group without pre-training, we searched learning rate between 1e-4 to 1e-3 and found 2e-4 the best.

**Results** Main results are provided in Table 1. Due to the choice of acoustic feature, our method suffers from the imperfect pseudo-inverse of Mel filters and the lack of phase modeling. In contrast to prior works tailored for enhancement, these restrictions result in a worse upper-bound as shown in the table. Nevertheless, our method still provided comparable or better results against the prior works on both benchmark datasets. Despite using a dataset with different topics and speakers, generative pre-training still improved enhancement results compared to the same model trained on VB-DMD from scratch. Especially on the out-of-domain WSJ0-CHiME3 testing, SpeechFlow demonstrated strong generalizability with a clear gap on PESQ, CSIG, and COVL against all other methods. In the case where the larger dataset DNS2020 is used for fine-tuning, a similar trend can be found compared to prior work DEMUCS and the testing result on WSJ0-CHiME3 can be further improved. These results pointed out the great potential of generative pre-training on speech.

### 4.3 FINE-TUNING FOR SPEECH SEPARATION

**Task & Metrics** The goal of separation is to separate mixture (overlapped) speech into multiple single-speaker speech. In our experiment, we focus on separating 2 to 3 speakers for simplicity. We report the common metric Scale-Invariant Signal-to-Distortion Ratio improvement (SI-SDRi; Le Roux et al., 2019) that measures the improvement of separated speech over the mixture when comparing against the clean reference in the time domain. In addition, we also report the ESTOI improvement (ESTOIi) of the separation result over the mixture to measure the intelligibility.

**Dataset & Prior Work** For separation, SpeechFlow is fine-tuned using a synthetic mixture created by randomly sampling and mixing 2 or 3 utterances from 360 hours of speech from English audiobook. In addition, noise sampled from WHAM! dataset (Wichern et al., 2019) can be added to the mixture to further increase the difficulty of separation, combining 4 different setups in total. We tested the fine-tuned model on LibriMix (Cosentino et al., 2020) 16khz min. For training, paired data $(x_1^1, x_1^2, y)$ is provided where $x_1^1, x_1^2$ are the target clean signal and $y$ is the mixture. Signals

Table 3: English zero-shot speaker adaptation TTS results on filtered LS (Panayotov et al., 2015) test-clean. Best results are **bolded**. For cross-sentence reference, the speaker information is provided by a 3-second prompt from a different utterance sampled randomly. For continuation, the first 3 seconds of the target utterance is used. FT stands for fine-tuning the full model; LoRA stands for fine-tuning with Low-rank Adaptors (Hu et al., 2021) where pre-trained weights are frozen.

| Method | labeled data (hr) | cross-sentence reference | | | continuation | | | subjective MOS |
|---|---|---|---|---|---|---|---|---|
| | | WER | SIM-o | SIM-r | WER | SIM-o | SIM-r | |
| Ground truth | | - | - | - | 2.2 | 0.754 | - | 3.80 |
| YourTTS (Casanova et al., 2021) | 475 | 7.7 | 0.337 | n/a | - | - | - | 2.92 |
| VALL-E (Wang et al., 2023) | 60k | 5.9 | - | 0.580 | 3.8 | 0.452 | 0.508 | - |
| Voicebox (Le et al., 2023) | 60k | **1.9** | 0.662 | 0.681 | **2.0** | 0.593 | 0.616 | 3.54 |
| Single GPU training | | | | | | | | |
|   SpeechFlow w/o pre-train | 960 | 2.3 | 0.526 | 0.573 | 2.2 | 0.467 | 0.513 | - |
|   SpeechFlow FT | 960 | 2.2 | 0.678 | 0.694 | 2.2 | 0.613 | 0.630 | - |
|   SpeechFlow LoRA | 960 | 2.6 | 0.696 | 0.711 | 2.4 | 0.623 | 0.640 | - |
| 32 GPU training | | | | | | | | |
|   SpeechFlow w/o pre-train | 960 | 2.0 | 0.569 | 0.598 | 2.1 | 0.530 | 0.557 | - |
|   SpeechFlow FT | 960 | 2.2 | 0.697 | 0.703 | 2.2 | 0.622 | 0.629 | - |
|   SpeechFlow LoRA | 960 | 2.1 | **0.700** | **0.715** | 2.1 | **0.630** | **0.644** | 3.43 |

are randomly cropped into 8-second chunks for training. To ensure the model outputs all speakers, we concatenated the clean signals along the time axis (and repeated the condition $y$ accordingly) for both training and testing. The baseline system is Conv-TasNet (Luo & Mesgarani, 2019) from LibriMix[1]. We note that while there are many other prior works in the field, most of them focused on WSJ2mix dataset (Hershey et al., 2016) with 8kHz audio, which makes fair comparison difficult. To provide a more competitive baseline, we reproduce a more powerful separation model SepFormer (Subakan et al., 2021; 2023) at 16kHz using code provided by the authors [2].

**Training** The fine-tuning setup follows enhancement with few changes: batch size is reduced to 37.5 seconds; model is fine-tuned for 85 epochs; peak learning rate is set to 3e-5. For Speech-Flow without pre-training, we searched learning rate between 1e-5 to 1e-4 and found 5e-5 the best.

**Results** Results are provided in Table 2. We found SI-SDRi more sensitive to the process of Mel-spectrogram-to-waveform. This can be verified by examining the upper-bound performance using a clean reference Mel spectrogram, which is even worse than the baseline Conv-TasNet. Similarly, we found the more recent transformer-based model SepFormer (Subakan et al., 2023) struggled in SI-SDRi when training at 16kHz (i.e., 2x longer input). In contrast, we found ESTOIi that reflected the intelligibility of separation result more robust to waveform estimation. Nevertheless, fine-tuned SpeechFlow was able to provide strong separation results. The gap between SpeechFlow and its upper-bound is particularly small in the easy 2 Mix setup. To measure the true quality of the Mel spectrogram generated by SpeechFlow, we also experimented with learnable inverse-Mel and phase estimation (as described in Section A.2) and found the separation result can be further boosted in terms of SI-SDRi. Since optimizing the Mel-spectrogram-to-waveform transform is beyond the scope of this paper, we apply learnable estimation to the best result of 2 Mix and 2 Mix + Noise only. The key idea is to show the separation result in the Mel spectrogram is already at a high quality, and metrics that are limited by the choice of input/output feature like SI-SDRi can be further improved with extra effort. In conclusion, we found SpeechFlow providing better intelligibility in all cases. It is worth noting that the fine-tuning method presented here is a vanilla solution that might not scale well as the number of speakers increases, a more dedicated fine-tuning method is left as future work.

## 4.4 FINE-TUNING FOR ZERO-SHOT SPEAKER ADAPTATION OF TEXT-TO-SPEECH

**Task & Metrics** We consider speech generation conditioning on text, i.e., text-to-speech (TTS). In particular, we focus on the zero-shot speaker adaptation problem (Jia et al., 2018; Casanova et al., 2021) where the voice of an unseen speaker should be used for synthesis. The problem

---

[1] https://huggingface.co/JorisCos

[2] https://github.com/speechbrain/speechbrain/tree/v0.5.15/recipes/LibriMix

setup and the evaluation metrics followed VALL-E (Wang et al., 2023) and Voicebox (Le et al., 2023). Zero-shot adaptation is done by using a 3-second prompt that carries speaker, paralinguistic, and environmental information. To measure the correctness and the intelligibility of the synthetic speech, we measure the recognition word error rate (WER) using HuBERT-L (Hsu et al., 2021) pre-trained and fine-tuned on LibriLight (Kahn et al., 2019) and LibriSpeech (Panayotov et al., 2015) respectively. Using WavLM-TDCNN speaker embedding model Chen et al. (2022), speaker similarity is measured by the similarity between the embedding of generated speech and that of the conditioning audio. Similarity to the original conditioning audio (SIM-o) and to the vocoder-resynthesized audio (SIM-r) are reported. In addition to the objective metrics, subjective evaluation on cross-sentence reference results using mean opinion score is also provided. See more detail regarding MOS test in Section A.4.6.

**Prior Works** YourTTS (Casanova et al., 2021) is a flow-based model (Kim et al., 2021) trained on multi-lingual data, including VCTK (Yamagishi et al., 2019), TTS-portugese (Casanova et al., 2022), M-AILABS French (Munich Artificial Intelligence Laboratories GmbH, 2017), and LibriTTS (Zen et al., 2019). VALL-E is a decoder-only auto-regressive model trained on LibriLight for zero-shot speaker adaptation TTS. Lastly, the closely related prior work Voicebox combined flow-matching and masked prediction for supervised TTS training. Voicebox can be viewed as a strong baseline using the same amount of data with fully supervised training.

**Dataset** 960 hours of transcribed speech from English audiobook is used for fine-tuning. The testing protocol follows VALL-E and Voicebox. Montreal Force Aligner (McAuliffe et al., 2017) is used for phone-speech alignment. Position postfixes are added to each phone following Voicebox. Additional results on fine-tuning with less (100/10 hours) labeled data are provided in Section A.4.4.

**Training** To enable zero-shot speaker adaptation , fine-tuning condition $y$ includes masked audio $x_m$ and the force-aligned phone sequence. We followed the masking strategy of Voicebox during fine-tuning. We additionally tested fine-tuning with more (32) GPUs and Low-rank Adaptors (LoRA; Hu et al., 2021; we use rank $r = 64$) to study the impact of computational resource for fine-tuning. Section A.4.2 provided a detailed performance analysis based on the number of GPUs used for fine-tuning. The batch size is 75 seconds per GPU in all cases. For standard fine-tuning, the learning rate is set to peak at 1e-5 after 5k updates, then linearly decay to 0 for the rest 145k steps. For LoRA fine-tuning, 9.5M new learnable parameters are introduced to the pre-trained model, accounting for 2.8% of the full model. All pre-trained weights are frozen. The learning rate is set to peak at 1e-3. Additional results on the impact of the amount of fine-tuning GPU is provided in Section A.4.3 .

**Results** Results are provided in Table 3. Comparing to fully supervised models Voicebox or VALL-E, a clear advantage in speaker modeling can be found with SpeechFlow despite using much less labeled data. In terms of WER and MOS, SpeechFlow is slightly worse than Voicebox that uses more labeled data. In addition, while single GPU fine-tuning already provided better speaker adaptation than all baselines, we found fine-tuning with more GPUs provided even stronger results. Interestingly, LoRA performed the best in terms of both SIM and WER among all fine-tuning setups. This suggested that fine-tuning method for generative model could be worth exploring in the future. Finally, our baseline without pre-training achieved similar WER to that of the pre-trained model but a significantly worse SIM. These findings suggested the proposed generative pre-training improves speaker modeling but not content modeling for speech synthesis.

## 4.5 MULTI-TASK FINE-TUNING OF SPEECHFLOW

Preceding sections showed SpeechFlow can be fine-tuned for different purpose using limited paired data and/or computation. In this section we take one step further to investigate the possibility to build an all-in-one controllable speech generation model via multi-task fine-tuning. Results are carried out in Table 4. We simply combined the labeled datasets for enhancement (DNS), separation (2Mix+Noise), and TTS for fine-tuning. We upsampled these datasets with a factor of 10/4/1 respectively to balance the importance of each task. Pre-trained SpeechFlow is fine-tuned on single GPU for 700k updates with the same learning rate scheduler peaking at 2e-5.

Table 4: Results for multi-task fine-tuning. Both single-task and multi-task SpeechFlow are fine-tuned using single GPU. Expert models are the best prior work for each metric of each task from Table 1,2,3. For TTS /enhancement/separation, we consider the cross-reference/WSJ0-CHiME3/2Mix+Noise scenario respectively. ZSSA is short for zero-shot speaker adaptation.

| Method | ZSSA TTS | | Enhancement | | Separation | |
|---|---|---|---|---|---|---|
| | WER | SIM-o | PESQ | COVL | SI-SDRi | ESTOIi |
| Single-task models | | | | | | |
|    Expert prior work | **1.9** | 0.662 | 2.49 | 3.32 | **12.55** | 0.28 |
|    SpeechFlow | 2.2 | **0.678** | **2.87** | 3.54 | 12.41 | **0.37** |
| Multi-task models | | | | | | |
|    SpeechFlow | 2.3 | 0.651 | **2.87** | **3.56** | 9.73 | 0.30 |

For zero-shot speaker adaptation TTS, we observed a drop on both WER and SIM-o, suggesting multi-task learning can lead to worse performance in specific single task. However, multi-task results are found to be better than single-task ones for enhancement. One possible explanation is the separation task trained on mixture+noise can also be viewed as a hard enhancement problem the model was additionally trained on. This showcased the benefit of having a universal model - some tasks might benefit from others. For separation, we found multi-task model deteriorated significantly comparing to the single task model. Preliminary results presented in this section suggested an all-in-one speech generative model can be built from SpeechFlow, but further research and development is required to improve the results and cover a more diverse set of tasks.

## 5 CONCLUSION

In this paper, we studied the role of generative model as a foundation model instead of a tool for a specific task. We show that training SpeechFlow using flow matching with masked condition results in a strong generative model. The model can be deployed to different downstream tasks using simple fine-tuning strategy with a single GPU. In our experiment, we adapted SpeechFlow to speech enhancement, separation, and zero-shot speaker adaptation TTS with performance comparable to task-specific models. More importantly, SpeechFlow demonstrated the potential to unify generative tasks for speech.

**Limitations and Future Works** This work focused on developing the pre-train-and-fine-tune framework for generative speech model. For the selected downstream applications, we assumed a frame-wise condition (e.g., noisy spectrogram; force-aligned phone label) is available in the fine-tune dataset. Fine-tuning with misaligned data (e.g., raw text, speaker ID) is left as an important future work. In addition, SpeechFlow is trained and tested on English-only data. However, since the generative model can be trained without label data, we believe the method can be easily scaled to more languages in the future. For future works, we would like to point out that the choice of acoustic feature may limit the applications as we discovered in enhancement and separation. Hence finding a more general acoustic feature would be a key step to general purpose generative speech model. Finally, we note that some of the expert models compared in different downstream tasks have other focuses besides the reported metrics (e.g., DEMUCS is built to run in real-time with fewer parameters). Therefore, we would like to emphasize that this work is mainly to show the potential of pre-trained generative models rather than claiming state-of-the-art in different tasks.

## ACKNOWLEDGMENTS

The authors would like to thank Gene-Ping Yang for helpful discussions on speech separation and Baishan Guo for setting up human evaluation.

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

# A  APPENDIX

## A.1  AUDIO SAMPLES

Audio samples can be found at `https://voicebox.metademolab.com/speechflow.html`. For additional samples, please refer to the supplementary materials at `https://openreview.net/forum?id=KpoQSgxbKH`.

## A.2  INFERENCE DETAILS

**Generating Mel Spectrogram**    To generate Mel spectrogram $x_1$, we first sample $x_0$ from the simple prior $p_0(x)$. The next step is to estimate $\phi_1(x_0)$ given $\phi_0(x_0) = x_0$ by evaluating $v_t(\phi_t(x_0), y; \theta)$ at multiple $t$. Each evaluation required forwarding through the neural network, and a larger number of function evaluations (NFEs) leads to a more accurate estimation of $\phi_1(x_0)$. In addition, we also applied classifier-free guidance (CFG; Dhariwal & Nichol, 2021; Le et al., 2023) to prioritize audio quality over diversity. CFG is done by additionally predicting the unconditioned vector field $v_t(\phi_t(x_0); \theta)$ (where the task-specific condition is dropped) to obtain a modified prediction

$$\tilde{v}_t = (1 + \alpha) \cdot v_t(\phi_t(x_0), y; \theta) + \alpha \cdot v_t(\phi_t(x_0); \theta). \tag{6}$$

CFG allows us to improve sample quality by focusing more on task-specific conditioned generation with larger $\alpha$ at the cost of doubling NFEs. We use $\alpha = 0.5$ for enhancement and $0.7$ for other tasks in practice. For the ODE solver, we use midpoint method implemented in `torchdiffeq` (Chen, 2018) to derive $\phi_1(x_0)$ from $\phi_0(x_0)$ by approximating the integration from $t = 0$ to $t = 1$ with a step size of 0.0625, resulting 32 NFEs per sample.

**Zero-shot Speaker Adaptation TTS**    To generate audible speech from Mel spectrogram, HiFi-GAN vocoder (Kong et al., 2020) from VoiceBox (Le et al., 2023) is adopted. In addition, phone duration is also needed to determine the output spectrogram length and the frame-wise condition given the input phone sequence. The regression-based duration predictor from VoiceBox is adopted for all TTS-related experiments.

**Speech Enhancement**    Different from TTS, enhancement metrics are more sensitive to the sample-to-sample alignment of the waveform between the hypothesis and reference. This makes Neural vocoder a bad option for the task[3]. Alternatively, we found using pseudo-inverse of Mel filter bank to recover linear Spectrogram, adding phase information taken directly from the noisy speech (input condition), and apply inverse Short-Time Fourier Transform (iSTFT) sufficient[4]. As a reference, PESQ on WSJ0-CHiME3 dropped from 2.70 to 2.29 when switching the signal processing method to HiFi-GAN vocoder.

**Speech Separation**    For this task, we found both the signal processing method and HiFi-GAN vocoder not enough for the most popular metric SI-SDRi (see discussion in Section 4.3). To this end, we train a 3-layer ResNet for both pseudo-inverse Mel transform and phase estimation using precomputed Mel spectrogram prediction and target waveform on the training set. The model takes the separation result (Mel spectrograms from SpeechFlow) and the complex spectrogram of the mixture as input, predicting both the linear spectrogram and the phase information to be combined and transformed to the time domain with iSTFT. Since the whole process is differentiable, the model is trained to maximize the permutation-invariant (Yu et al., 2017) SI-SDR loss against the target waveform.

---

[3] See the last section in demo page for examples.

[4] See the topline section in Table 7 for the error introduced by the process.

## A.3 MODEL ARCHITECTURE AND CONDITION DETAILS

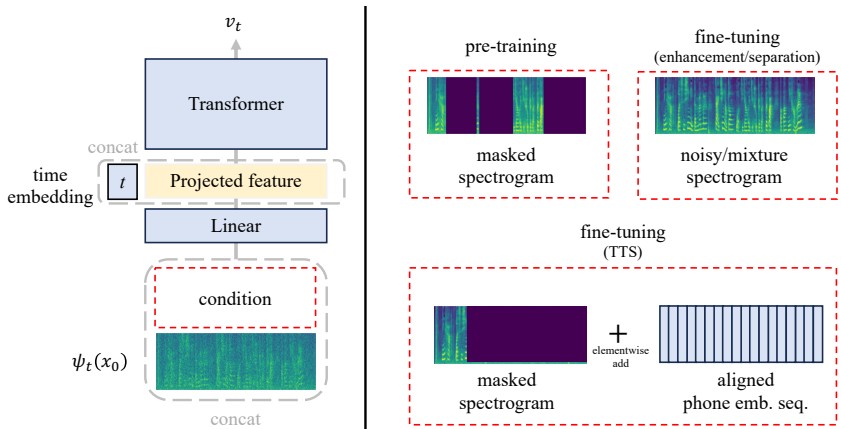

Figure 2: Blue blocks are learnable weights. (Left) Model architecture. Time (flow step) $t$ is encoded using sinusoidal position embedding with learnable scale. (Right) Condition used for different tasks. For TTS fine-tuning, learnable phone embedding sequence aligned to the spectrogram is element-wise added to the masked spectrogram. Since phone embeddings are randomly initialized and added to the masked spectrogram, we found ramping up a zero-initialized gating value (single scalar to be multiplied on phone embedding) yields slightly better results in practice.

Table 5: Detailed configurations for training. ConvPos stands for Convolutional Positional Embedding (Baevski et al., 2020); Skip Connections are introducted in Le et al. (2023); Alibi Bias is introducded in Press et al. (2021).

| | Pre-training | Fine-tuning | | |
| --- | --- | --- | --- | --- |
| | | Enhancement | Separation | TTS |
| | | Model Parameters | | |
| Model Dimension | | | 1024 | |
| Number of Heads | | | 16 | |
| Number of Layers | | | 24 | |
| Feedforward Dimension | | | 4096 | |
| Attention Dropout | | | 0.0 | |
| Activation Dropout | | | 0.1 | |
| ConvPos Width | | | 31 | |
| ConvPos Groups | | | 16 | |
| ConvPos Depth | | | 2 | |
| Skip Connections | | | true | |
| Alibi Bias | | | true | |
| Additional weights | - | No | No | 80-dim. phn. emb. |
| | | Hyper-Parameters | | |
| Condition drop rate $p_{drop}$ | 10% | 30% | 30% | 20% |
| Masking probability $n_{mask}$ | $\sim \mathcal{U}[70\%, 100\%]$ | 0% | 0% | $\sim \mathcal{U}[70\%, 100\%]$ |
| Minimum mask span $l_{mask}$ | 10 frames | N/A | N/A | 10 frames |
| | | Training Parameters | | |
| Number of Updates | 600k | (dataset-dependent, see Section 4.2,4.3) | | 150k |
| Number of GPUs | 32 | 1 | 1 | 1 to 32 |
| Batchsize per GPU | 75 seconds | 50 seconds | 37.5 seconds | 75 seconds |
| Max length per audio | | 16 seconds | | |
| Learning Rate | 5e-5 | 2e-5 | 3e-5 | 1e-5 |
| Gradient Clipping Value | | 0.2 | | |
| LR Scheduler Warmup Steps | | 5000 | | |

## A.4 ADDITIONAL RESULTS

### A.4.1 SPEECH EDITING

Following the setup of A3T (Bai et al., 2022), here we additionally consider the task where the center 50% of an recording is to be edited. See Figure 3 and Section 4.4 in Bai et al. 2022 for more details and illustration of the task. Similar to A3T and Voicebox (Le et al., 2023) that are trained with masked audio and text conditioning, speech editing is another downstream that can be naturally solved with the fine-tuned SpeechFlow. Results are presented in Table 6, evaluation metrics and the dataset follows the zero-shot speaker adaptation TTS experiment presented in Section 4.4. Similar to zero-shot speaker adaptation TTS, we found SpeechFlow performing close to the state-of-the-art model using much less labeled data thanks to pre-training.

Table 6: Speech editing results on filtered LS (Panayotov et al., 2015) test-clean. Please refer to Section 4.4 for the metrics used here.

| Method | labeled data (hr) | WER | SIM-o |
|---|---|---|---|
| A3T (Bai et al., 2022) | 44 | 11.5 | 0.148 |
| Voicebox (Le et al., 2023) | 60k | **2.0** | 0.613 |
| SpeechFlow LoRA | 960 | 2.2 | **0.647** |

### A.4.2 FULL RESULT FOR SPEECH ENHANCEMENT

Table 7: Speech enhancement results on the test set of Voicebank-Demand (Valentini-Botinhao et al., 2017) and WSJ0-CHiME3 (Richter et al., 2023). All metrics are the higher the better, best result of each section is **bolded**. Numbers are taken from prior works unless otherwise specified. PSEQ-nb is the narrow-band version of PSEQ. CBAK refers to composite background intrusiveness (Hu & Loizou, 2007).

| Method | Voicebank-Demand | | | | | | WSJ0-CHiME3 | | | | | |
|---|---|---|---|---|---|---|---|---|---|---|---|---|
| | PESQ | PESQ-nb | ESTOI | CSIG | CBAK | COVL | PESQ | PESQ-nb | ESTOI | CSIG | CBAK | COVL |
| Baseline | | | | | | | | | | | | |
| Mixture | 1.97 | 2.88 | 0.79 | 3.35 | 2.44 | 2.63 | 1.69 | 2.29 | 0.78 | 3.24 | 2.26 | 2.42 |
| Models trained on Voicebank-Demand | | | | | | | | | | | | |
| SGMSE+ (Richter et al., 2023) | 2.93 | 3.66 | **0.87** | 4.13[*] | 3.39[*] | 3.53[*] | 2.48 | 3.12[*] | **0.90** | 3.67[*] | 2.95[*] | 3.02[*] |
| Conv-TasNet[†](Luo & Mesgarani, 2019) | 2.63 | 3.42 | 0.85 | - | - | - | 2.40 | - | 0.88 | - | - | - |
| MetricGAN+ (Fu et al., 2021) | **3.13** | 3.63 | 0.83 | 4.10[*] | 2.90[*] | 3.61[*] | 2.13 | 2.67[*] | 0.76 | 3.02[*] | 1.88[*] | 2.52[*] |
| DEMUCS (Défossez et al., 2020) | 3.07 | - | - | 4.31 | 3.40 | 3.63 | - | - | - | - | - | - |
| SpeechFlow | **3.13** | **3.74** | **0.87** | **4.43** | **3.41** | **3.80** | **2.70** | **3.36** | **0.90** | **4.05** | **2.97** | **3.36** |
| SpeechFlow w/o pre-train | 2.92 | 3.57 | 0.85 | 4.22 | 3.26 | 3.57 | 2.38 | 3.02 | 0.86 | 3.72 | 2.75 | 3.03 |
| Models trained on Deep Noise Supression Challange 2020 (Reddy et al., 2020) | | | | | | | | | | | | |
| DEMUCS (Défossez et al., 2020) | 2.55[*] | 3.40[*] | 0.85[*] | 3.24[*] | 3.26[*] | 2.88[*] | 2.49[*] | 3.20[*] | **0.92**[*] | 3.93[*] | **3.24**[*] | 3.20[*] |
| SpeechFlow | **2.71** | **3.65** | 0.86 | **4.07** | 2.93 | **3.39** | **2.87** | **3.45** | 0.91 | **4.24** | 3.14 | **3.54** |
| Topline | | | | | | | | | | | | |
| Ours upper-bound[‡] | 3.77 | 4.09 | 0.95 | 4.97 | 4.00 | 4.54 | 3.68 | 3.93 | 0.96 | 4.97 | 3.81 | 4.46 |
| Clean signal | 4.50 | 4.55 | 1.00 | 5.00 | 5.00 | 5.00 | 4.50 | 4.55 | 1.00 | 5.00 | 5.00 | 5.00 |

[*] Results reproduced by us using the open sourced model released by the authors.
[†] Results reproduced by Richter et al. (2023).
[‡] Obtained from Mel Spectrogram of the clean signal, error introduced by pseudo-inversing Mel filter bank and taking phase from the mixture.

### A.4.3 Increasing the Number of GPUs for Fine-tuning

Unsurprisingly, more GPUs (larger batch size) results in better performance in general. Given the fact that fine-tuning have smaller gap between the result of using 1 and 32 GPUs, it is worth noting that fine-tuning is more robust than training from scratch in terms of speaker similarity.

Table 8: Additional results of English zero-shot speaker adaptation TTS experiment with different number of GPUs. 960 hours of labeled data is used.

| # GPUs | cross-sentence reference | | | continuation | | |
|---|---|---|---|---|---|---|
| | WER | SIM-o | SIM-r | WER | SIM-o | SIM-r |
| LoRA (Hu et al., 2021) fine-tuning | | | | | | |
| 1 (default) | 2.6 | 0.696 | 0.711 | 2.4 | 0.623 | 0.640 |
| 2 | 2.5 | 0.695 | 0.710 | 2.3 | 0.623 | 0.640 |
| 4 | 2.4 | 0.698 | 0.713 | 2.2 | 0.623 | 0.639 |
| 8 | 2.3 | 0.697 | 0.712 | 2.2 | 0.623 | 0.639 |
| 16 | 2.2 | 0.697 | 0.712 | 2.2 | 0.625 | 0.641 |
| 32 | 2.1 | 0.700 | 0.715 | 2.1 | 0.630 | 0.644 |
| Training from scratch | | | | | | |
| 1 | 2.3 | 0.526 | 0.573 | 2.2 | 0.467 | 0.513 |
| 32 | 2.0 | 0.569 | 0.598 | 2.1 | 0.530 | 0.557 |

### A.4.4 Reducing Labeled Data for Fine-tuning

Interestingly, we found pre-trained model generalized better to unseen speaker comparing against models trained from scratch. However, it is also harder to overfit the pre-trainiend model on the limited amount of text input, resulting a worse intelligibility in terms of WER. Nevertheless, with 10 hours of fine-tuning data, SpeechFlow was able to outperform VALL-E (Wang et al., 2023) that was trained on 60k hours data.

Table 9: Additional results of English zero-shot speaker adaptation TTS experiment using less labeled data. Single GPU is used for fine-tuning the whole pre-trained model.

| Method | labeled data (hr) | cross-sentence reference | | | continuation | | |
|---|---|---|---|---|---|---|---|
| | | WER | SIM-o | SIM-r | WER | SIM-o | SIM-r |
| Ground truth | | - | - | - | 2.2 | 0.754 | - |
| YourTTS (Casanova et al., 2021) | 475 | 7.7 | 0.337 | n/a | - | - | - |
| VALL-E (Wang et al., 2023) | 60k | 5.9 | - | 0.580 | 3.8 | 0.452 | 0.508 |
| Voicebox (Le et al., 2023) | 60k | 1.9 | 0.662 | 0.681 | 2.0 | 0.593 | 0.616 |
| SpeechFlow w/o pre-train | 960 | 2.3 | 0.526 | 0.573 | 2.2 | 0.467 | 0.513 |
| | 100 | 2.3 | 0.412 | 0.463 | 2.2 | 0.370 | 0.417 |
| | 10 | 2.4 | 0.360 | 0.410 | 2.3 | 0.330 | 0.374 |
| SpeechFlow | 960 | 2.2 | 0.678 | 0.694 | 2.2 | 0.613 | 0.630 |
| | 100 | 2.8 | 0.613 | 0.632 | 2.5 | 0.555 | 0.573 |
| | 10 | 4.1 | 0.578 | 0.600 | 3.1 | 0.520 | 0.541 |

## A.4.5 IMPACT OF PRE-TRAINING HYPER-PARAMETER

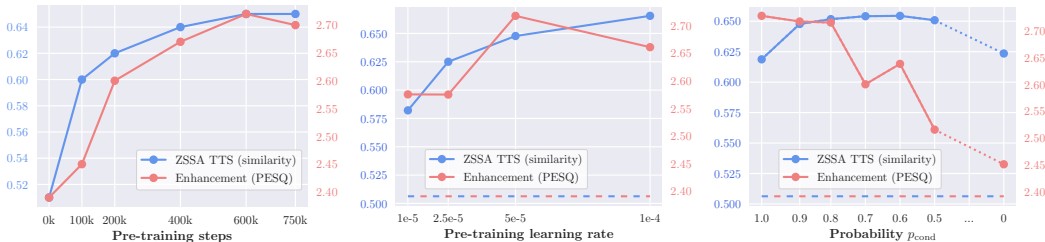

Figure 3: Impact of different pre-training hyper-parameters on zero-shot speaker adaptation (ZSSA) TTS and enhancement. The dashed line stands for the baseline performance without pre-training.

Since the the main focus of our method is on pre-training generative speech model, we provide study on the corresponding hyper-parameters here. To evaluate the pre-trained model in a less biased perspective, we consider both speaker similarity of zero-shot speaker adaptation TTS and PESQ of enhancement for multi-task fine-tuned model. Results are provided in Figure 3.

**Learning Rate & Number of Updates.** First, we investigate the pre-trained model quality as a function of the number pre-training steps or learning rate. We set the total number of updates to 750k, which is about 7 epochs on training set. One caveat is that learning rate decay is applied through out the training, which could also contribute to the tapering result. We found most of the gain coming from the early stage before 400 updates and setting the learning rate above 5e-5 is sufficient for stable result.

**Conditioning.** We also found the model to be stable when setting $p_{cond}$ above $80\%$. Importantly, we also found unconditioned pre-training, i.e., $p_{cond} = 0$, yielded bad performance on both tasks. The result showcased the helpfulness and the necessity of masked prediction for pre-training. In summary, SpeechFlow is stable as long as masked conditioning is prioritized (over unconditioned pre-training) and the model is trained with sufficient steps and step size.

**Masking Hyper-parameter.** Table 10 studied the impact of the proportion for placing mask $n_{mask}$ and the masking span size $l_{mask}$. In simple terms, we found masking a significant proportion is important for SpeechFlow.

Table 10: Additional results of English zero-shot speaker adaptation TTS experiment with different pre-training hyper-parameters. To reduce computation, models in this table are only pre-trained for 300k steps.

| | cross-sentence reference | | | continuation | | |
|---|---|---|---|---|---|---|
| | WER | SIM-o | SIM-r | WER | SIM-o | SIM-r |
| $n_{mask} \sim \mathcal{U}[70\%, 100\%], l_{mask} = 10$ (default) | 2.2 | 0.655 | 0.669 | 2.1 | 0.596 | 0.610 |
| $n_{mask} \sim \mathcal{U}[80\%, 100\%]$ | 2.2 | 0.627 | 0.644 | 2.1 | 0.580 | 0.597 |
| $n_{mask} \sim \mathcal{U}[60\%, 100\%]$ | 2.3 | 0.581 | 0.592 | 2.1 | 0.562 | 0.574 |
| $n_{mask} \sim \mathcal{U}[60\%, 90\%]$ | 2.2 | 0.599 | 0.612 | 2.1 | 0.567 | 0.577 |
| $n_{mask} \sim \mathcal{U}[70\%, 90\%]$ | 2.1 | 0.614 | 0.623 | 2.1 | 0.589 | 0.596 |
| $l_{mask} = 5$ | 2.0 | 0.661 | 0.677 | 2.1 | 0.600 | 0.616 |
| $l_{mask} = 15$ | 2.2 | 0.609 | 0.629 | 2.2 | 0.585 | 0.605 |

A.4.6  SUBJECTIVE EVALUATION FOR ZERO-SHOT SPEAKER ADAPTATION TTS

Table 11: Subjective test on English zero-shot speaker adaptation TTS on filtered LS test-clean with cross-sentence reference. Averaged rating along with 95% confidence interval are reported for Mean Opinion Score (MOS).

| Method | labeled data (hr) | objective metrics | | | subjective MOS |
| --- | --- | --- | --- | --- | --- |
| | | WER | SIM-o | SIM-r | |
| Ground truth | - | - | - | - | 3.80±0.09 |
| YourTTS (Casanova et al., 2021) | 475 | 7.7 | 0.337 | n/a | 2.92±0.10 |
| Voicebox (Le et al., 2023) | 60k | 1.9 | 0.662 | 0.681 | 3.54±0.08 |
| SpeechFlow | 960 | 2.1 | 0.700 | 0.715 | 3.43±0.09 |

In addition to objective metrics that covers intelligibility and similarity measured by models, we conducted human evaluation to measure the overall quality of audio samples using Mean Opinion Score (MOS) following CrowdMOS (Ribeiro et al., 2011). We randomly selected 50 sentences from the LS test-clean for human evaluation. Each audio sample received 10 ratings in total. Each participant was asked to rate 20 audio samples, including 5 different sentences with audio from 4 different sources - ground truth, YourTTS (Casanova et al., 2021), Voicebox (Le et al., 2023), and SpeechFlow. Results are collected through Amazon Mechanical Turk (AMT) with task description provided in Table 12. Annotators are filtered with the following qualifications: (1) They need to be wearing a headset; (2) They need to pass an onboarding test (2 simple questions, where in each question people need to pick an audio with higher quality); (3) Post-processing, correlation coef between annotators' answer and the majority answer greater than 0.2.

From the MOS results in Table 11, we confirmed that SpeechFlow is able to generate high quality audio judging by human, falling only slightly behind its fully supervised counterpart Voicebox while using over 62.5x less labeled data.

Table 12: Mean opinion score (MOS) instruction.

**Introduction**
Hello! We need your help to evaluate the subjective quality and intelligibility of speech. In each task, you will evaluate a 2-8s speech segment and rate its overall quality from 1 to 5. You will be given 20 questions, and it will take you 5-10 minutes to finish.
(1) Please use a headset for listening and adjust your volume level to your comfort during this training, and do not change later during the experiment.
(2) Please consider the following aspects when evaluating the overall quality: (a) clarity of speech (b) sound quality (c) naturalness.
For each of the speech audio below, rate the overall speech quality on a scale from 1-5. (You need to play the speech audios in order to make a selection!)

**Score (Quality and Intelligibility of the speech)**
5 (Excellent)
4 (Good)
3 (Fair)
2 (Poor)
1 (Bad)

