# OpenReview forum: "Generative Pre-training for Speech with Flow Matching"
_ICLR.cc/2024/Conference — ICLR 2024 poster_

### Official Review · Reviewer_STf2 · 2023-10-31

**Soundness:** 3 good
**Presentation:** 3 good
**Contribution:** 3 good
**Rating:** 6
**Confidence:** 2

**Summary:**

The paper proposes to pretrain a flow-based model with unsupervised pre-training and supervised fine-tuning paradigms. The pre-trained generative model can be fine-tuned with task-specific data for speech enhancement, separation, and synthesis. According to the results across several benchmarks, the proposed models match or surpass existing expert models.

**Strengths:**

The paper explores a novel direction to pre-train a general-purpose generative model with unlabeled speech using flow-based models. The most similar work is Voicebox, a flow-based model with supervised slot-filling training, and the authors conduct details discussion and experimental comparisons to show the advantages. The pre-trained model can be finetuned to support various tasks such as speech enhancement, separation, and synthesis. The experiments are convincing.

**Weaknesses:**

One primary limitation of this work is the relatively limited range of supported task types. It would be beneficial for the authors to expand their support to include a wider variety of tasks, such as speech editing tasks, to further demonstrate the capabilities of their pre-trained models. By incorporating additional task types, the authors can provide a more comprehensive evaluation of the model's abilities and showcase its versatility across various domains. This would enhance the overall contribution and applicability of the proposed pre-training approach.

**Questions:**

See weakness

---

> ### Author Response · Authors · 2023-11-17
>
> We thank the reviewer for the constructive and explicit feedback that would definitely improve our work. We have included new results on speech editing in Section A4.1 in the revision, following closely to the prior work A3T [A]. We also note that with the three fundamental tasks supported by our model, there are potentially more variants that can be easily done in the future. E.g., target speaker extraction with audio prompt, text-guided speaker extraction via multi-task model, speech removal via removing enhancement result, etc.
>
> We thank the reviewer again for helping improve this work, and we hope our response has addressed your concern and enhanced the overall contribution.
>
> [A] A3T:  Alignment-Aware Acoustic and Text Pretraining for Speech Synthesis and Editing, Bai et al.

---

> > ### Comment · Reviewer_STf2 · 2023-11-18
> > **Response to Authors**
> >
> > Thanks. I have read the response and maintain the same score.

---

### Official Review · Reviewer_d1Li · 2023-10-31

**Soundness:** 4 excellent
**Presentation:** 4 excellent
**Contribution:** 4 excellent
**Rating:** 8
**Confidence:** 5

**Summary:**

This paper proposes pre-training a flow-based speech synthesizer using 60kh of untranscribed speech, then fine-tuning it for downstream tasks including speech enhancement, speech separation, and zero-shot TTS.  In any self-supervised paradigm, one must find a way to add labels during the fine-tuning process; here, that problem is solved by using masked spectrograms as pseudo-labels during pre-training, then replacing those with actual labels (noisy speech, mixed speech, or phone sequences) during fine-tuning.  The idea of using masked spectrograms to condition flow was also used in the VoiceBox flow synthesizer, but that paper did not include a self-supervised pre-training stage.

**Strengths:**

Generative pre-training for speech synthesis might have been first proposed in "Semi-Supervised Training for Improving Data Efficiency in End-to-end Speech Synthesis" by Chung et al., 2019.  Generative flow was used for vocoding in "Waveglow: A flow-based generative network for speech synthesis," and was used for TTS in "Flowtron: an Autoregressive Flow-based Generative Network for Text-to-Speech Synthesis" --- neither of those papers used the flow matching paradigm, instead they trained the flow networks using end-to-end training criteria only.  The combination of these two ideas (flow-based TTS and pre-trained TTS) was not proposed in any paper I can find.  The new contribution of this manuscript, the use of generative flow in a self-supervised pre-training stage, is an elegant idea that forms a strong theoretical paradigm, and that is supported by strong experimental results compared to challenging baselines.

**Weaknesses:**

By omitting key references, this paper seems to be suggesting that nobody has ever thought of using self-supervised pre-training for speech synthesis before, and it seems to be suggesting that nobody has ever used generative flow for speech synthesis before.  The manuscript should include references to key works in both areas, in order to more clearly articulate what is the actual contribution of the paper.

**Questions:**

The paper should better describe the history of (1) the use of self-supervised pre-training for speech enhancement and speech synthesis, and (2) the use of generative flow in speech synthesis.  I recommend the following references, but I think there may be others that I'm missing:

Self-supervised training for TTS:  Yu-An Chung, Yuxuan Wang, Wei-Ning Hsu, Yu Zhang and RJ Skerry-Ryan, "Semi-supervised training for improving data efficiency in end-to-end speech synthesis," ICASSP 2019, 6940-6944

Self-supervised training for speech enhancement: Yang, Shu-wen, Po-Han Chi, Yung-Sung Chuang, Cheng-I. Jeff Lai, Kushal Lakhotia, Yist Y. Lin, Andy T. Liu et al. "Superb: Speech processing universal performance benchmark." arXiv preprint arXiv:2105.01051 (2021), and other papers that submitted entries to the Superb challenge.

Generative flow for speech synthesis:

Ryan Prenger, Rafael Valle and Bryan Catanzaro, "Waveglow: A flow-based generative network for speech synthesis," ICASSP 2019, 3617-3621

Rafael Valle, Kevin Shih, Ryan Prenger and Bryan Catanzaro, "Flowtron: an Autoregressive Flow-based Generative Network for Text-to-Speech Synthesis," arXiV 2020

**Details Of Ethics Concerns:**

The bibliography feels like it was filtered to remove papers written by authors at companies competing with the company at which these authors work.

---

> ### Author Response · Authors · 2023-11-17
>
> We would like to first thank the reviewer for the effort in providing a detailed review. Please see the following responses to the concerns/questions raised in the review and let us know if there are further questions.
>
> ---
>
> > By omitting key references, this paper seems to be suggesting that nobody has ever thought of using self-supervised pre-training for speech synthesis before. The manuscript should include references to key works in both areas, in order to more clearly articulate what is the actual contribution of the paper.
>
>
> We would like to highlight that the key innovation of our approach: SpeechFlow is, to the best of our knowledge, the first generative speech pre-training method that is not conditioned for any specific task (i.e., not just for speech synthesis, but also other generation tasks). We apologize if the writing is misleading in the submission and will do our best to avoid overstating our contribution.
>
> > The paper should better describe the history of (1) the use of self-supervised pre-training for speech enhancement and speech synthesis; …
>
> We thank the reviewer for pointing out the reference in semi-supervised TTS [Chung et al. 2019], we have updated Section 2 in our revision accordingly.
>
> For enhancement, the generative tasks are covered in related work (Section 2, under the “Pre-trained Speech Models” section) with references including the one suggested by the reviewer. We would also like to point out that the enhancement results in Superb-SG [A] are far behind recent enhancement models (that we compared against in this paper), results are more for benchmarking speech representation than building actual enhancement systems. This also showcased how SpeechFlow diverges from existing pre-trained ​​speech models – it is the first general-purpose pre-trained speech model for generative tasks.
>
>
> > … and it seems to be suggesting that nobody has ever used generative flow for speech synthesis before. …The paper should better describe the history of … (2) the use of generative flow in speech synthesis.
>
> The use of generative models in TTS has been discussed in Sections 1 and 2.  In particular, flow-based methods such as WaveGlow [Preger et at. 2019] are covered in the 1st paragraph of related work. We thank the reviewer for providing additional reference [Valle et at. 2020] for generative TTS, which is added in the revision.
>
> We would also like to point out that our approach relied on Flow Matching [B], which is different from flow-based methods. Flow matching learns a time-dependent diffeomorphic map, called “flow”, to push samples from one distribution to another in a continuous (as the mapping is defined by ODE function) manner. The neural network can be any model that predicts the trajectory of flow at any given time, and inference is done iteratively similar to diffusion models [C]. This is different from the flow-based generative models [D,E] which rely on strictly invertible networks to maximize the likelihood of data, and inference can be done in a single forward step.
>
> ---
>
> We thank the reviewer again for the constructive and helpful feedback, we hope the revised version addressed the concern raised in the review.
>
> ---
> ### Reference
> - [A] SUPERB-SG: Enhanced Speech processing Universal PERformance Benchmark for Semantic and Generative Capabilities, Tsai et al. 2022
> - [B] Flow matching for generative modeling, Lipman et al. 2022
> - [C] Denoising Diffusion Probabilistic Models, Ho et al. 2020
> - [D] NICE: Non-linear Independent Components Estimation, Dinh et al. 2014
> - [E] Glow: Generative Flow with Invertible 1×1 Convolutions, Kingma et al. 2018

---

### Official Review · Reviewer_C8mS · 2023-11-01

**Soundness:** 3 good
**Presentation:** 3 good
**Contribution:** 3 good
**Rating:** 6
**Confidence:** 4

**Summary:**

Speechflow is a generative model for speech generation and various tasks of it. It is trained with unlabeled speech with the goal of estimating the underlying distribution of speech conditioning on masked audio. Then it is fine-tuned for each specific task using labeled data.

**Strengths:**

-Novel idea on making a general-purpose speech generation model that can perform the following tasks outperforming the current SOTA approaches: speech enhancement, speech separation, zero-shot tts.
-I have listened to the audio samples and the model seems to perform well and produce high quality audio samples for all the tasks.
-Novelty of modeling speech directly.

**Weaknesses:**

-No subjective evaluation is presented which could be useful for the users of this model. Most TTS works present both subjective and objective metrics for the evaluation.
-The work is not a very good match for this venue. It would be more suitable in a speech-related venue like ICASSP or InternSpeech.

**Questions:**

-What dataset are you using? for pre-training. You mention 60k hours of English speech.
-Have you tried different mask instead of filling with zeros?

**Details Of Ethics Concerns:**

It is a text-to-speech model that can be possibly misused for harmful purposes. That being said I think that it shouldn't be rejected for that reason from this venue.

---

> ### Author Response · Authors · 2023-11-17
>
> We would like to first thank the reviewer for the feedback. Please see the following responses to the concerns/questions raised in the review and let us know if there are further questions.
>
> ---
>
>
> > No subjective evaluation is presented which could be useful for the users of this model. Most TTS works present both subjective and objective metrics for the evaluation.
>
> We thank the reviewer for the helpful feedback that helps improve our evaluation.  We have updated the human evaluation for TTS in Section A.4.6 and Table 11 in the paper.
>
>
> > What dataset are you using? for pre-training. You mention 60k hours of English speech.
>
> For pre-training, we use 60k hours of unlabeled speech from English audiobooks in the public domain recorded by thousands of speakers, which is a comparable setup as [A].
>
> > Have you tried different mask instead of filling with zeros?
>
> Yes, we have tested using learnable mask embeddings and found no improvement over zeroing out masked inputs in the early stage of this work. Filling zero provides equal performance with fewer model parameters.
>
> ---
> ### Reference
>
> [A] Neural Codec Language Models are Zero-Shot Text to Speech Synthesizers, Wang et al.

---

### Official Review · Reviewer_dpH8 · 2023-11-05

**Soundness:** 3 good
**Presentation:** 2 fair
**Contribution:** 2 fair
**Rating:** 3
**Confidence:** 5

**Summary:**

This paper proposes to use flow matching for speech generation. Experiments are conducted for speech enhancement, speech separation, and TTS.

**Strengths:**

The idea of using flow matching for speech synthesis is sound.

**Weaknesses:**

1. The novelty is limited. It's basically applying flow matching to the speech synthesis problem.
2. The evaluation of the experimental results are weak. For speech generation, subjective human evaluation are expected, especially for TTS. Without such evaluation, the results are not persuasive.
3. There is a disconnection between the main claim and the experimental results. The experimental results show strong performance with using flow matching for speech synthesis in a pretraining and fine-tuning matter (finetuning was done with large datasets, e.g. 360 / 960 hours speech) . However, it's not clear that it's a results of "a foundational model". The experiments of SpeechFlow without pretraining is not persuasive because it uses the same model size, which likely leads to overfitting.
4. The description of the experiments are severely limited. For example, what datasets were used, and the details on the model architecture and hyperparameteres. I have a major concern on reproductivity.

**Questions:**

- Sec 4.1 -- what are. the 60k hours of English speech data for training?
- Sec 4.3 -- can you give details on the 360 hours of training data?
- Sec 4.4 -- what's the 960 hours of transcribed English speech used for fine-tuning?
- Sec 4.4. -- the term "zero-shot TTS" is improper because the model is trained with fully supervised data. the reference is improper either as there are earlier and more established works not cited.
- Sec 3.1 -- typo: "variational audio encoders" => "variational autoencoders"

---

> ### Author Response · Authors · 2023-11-17
> **Response to concerns regarding novelty/evaluation**
>
> (Reference will be appended to the end of the last response)
>
> ---
> > The novelty is limited. It's basically applying flow matching to the speech synthesis problem.
>
> While speech synthesis[1] is indeed one of the applications studied here, **this paper focused more on developing pre-training methods (via flow matching) for generative tasks in speech**. Speech synthesis is only part of the experiment to showcase the use of a pre-trained model, alongside speech enhancement and separation.
>
> Our approach is different from existing generative modeling methods for speech. Generative models have been useful in speech generation tasks, such as GAN-based vocoders, flow-based TTS, diffusion-based enhancement, etc. Nevertheless, these generative models are trained with task-specific conditioning (e.g., vocoders are trained to model waveform conditioning on spectrograms), limiting their generalizability. In contrast, we propose to pre-train a generative model without a predefined application and show that the resulting model can be applied to different downstream generation tasks with strong performance.
>
> In short, **pre-training general-purpose generative models for speech have not yet been explored to the best of our knowledge**. (To further justify our novelty, this is indeed listed as the biggest strength of our work according to all other reviewers.) Given the current review is as short as “The novelty is limited. It's basically applying flow matching to the speech synthesis problem”, please kindly provide more detailed feedback:
> If the reviewer thinks a better solution has been proposed before, please list it so we can discuss the differences.
> If the reviewer thinks the problem we studied (a pre-training method that benefits multiple speech generative tasks) is irrelevant, please state why so we can discuss it.
>
> Otherwise, we hope our responses have clarified the contribution of this paper and we respectfully ask the reviewer to re-evaluate its novelty.
>
> [1] We assume the reviewer refers to text-to-speech synthesis as “speech synthesis” following the convention in the field. It could be possible that the reviewer is referring to a more general set of tasks with speech as output. In that respect, our method is novel as 1) it is the first general-purpose pre-trained generative model as described above; 2) flow matching has not been used on speech generation tasks besides text-to-speech.
>
> ---
>
> > The evaluation of the experimental results are weak. For speech generation, subjective human evaluation are expected, especially for TTS. Without such evaluation, the results are not persuasive.
>
> We thank the reviewer for the helpful feedback that helps improve our evaluation. **We have updated the human evaluation for TTS in Section A.4.6 and Table 11 in the paper.**
>
> Unlike TTS where the optimal solution might not be unique and involves a certain degree of subjective judgment, speech enhancement and separation datasets provide clean reference that is the optimal solution. Following well-recognized recent works [A,B,C] in the field, we report objective metrics without subjective evaluation and we believe our result is significant.
>
> **In addition, we also provided audible samples in the attachment so anyone can judge the quality subjectively.** More importantly, those samples are not cherry-picked. To be more specific, here is how the samples on the demo page are drawn:
> - For speech enhancement, we rank all testing samples from easy to hard using PESQ, and show the exact sample at the 0/20/40/60/80/100th percentile rank.
> - For zero-shot TTS, we use all the recordings used by prior work [D] in their demo [E] to provide a side-by-side comparison.
>
> While evaluation for speech generation tasks is indeed hard, the evaluation metrics we presented all are examined and standardized by preceding works. We have also added subjective test results as suggested by the reviewer. We hope our results are now considered persuasive by the reviewer.
>
> ---

---

> ### Author Response · Authors · 2023-11-17
> **Response to concerns regarding experiment and reproducibility**
>
> > There is a disconnection between the main claim and the experimental results. The experimental results show strong performance with using flow matching for speech synthesis in a pretraining and fine-tuning matter (finetuning was done with large datasets, e.g. 360 / 960 hours speech) . However, it's not clear that it's a results of "a foundational model". The experiments of SpeechFlow without pretraining is not persuasive because it uses the same model size, which likely leads to overfitting.
>
>
> **Our main claim is that a pre-trained generative model can serve as the foundational model of different generation tasks with competitive performance and less fine-tuning resources.** This is justified by (1) comparing to strong expert models in each task; and (2) comparing to the same model (in terms of architecture/model size/training loss) that was not pre-trained. The trend is consistent on all three different tasks SpeechFlow is tested on, not just speech synthesis.
>
> It is widely known that one of the key benefits of pre-training is that one can train a bigger model with fewer labeled data through pre-training. This has been proven effective in not just discriminative speech models [F,G], but also in image [H,I] and text [J,K] foundational models. **These models are called “foundation models” because they have been trained with unlabeled data at scale and can be easily applied to different downstream tasks with minimum supervision, just like the proposed model.**
>
> To further alleviate the reviewer's concern that the baseline might be overfitting, we conducted an additional experiment where the baseline model size is reduced from 24-layer (commonly known as “Large”) Transformer Encoder to 12-layer (commonly known as “Base”). **We found the smaller baseline has little to no gain compared to the original baseline that the reviewer suspected to be overfitting.** Below we attach the results (setup/metric follows Table 3 of the paper):
>
> |   | |  | cross  |  |  | cont. |    |
> |---|---|---|---|---|---|---|---|
> |   |  labeled  speech (hr) | WER  | SIM-o  | SIM-r  | WER  | SIM-o  | SIM-r  |
> |  Voicebox [D] |   60k | 1.9  | 0.662  | 0.681  | 2.0  | 0.593 | 0.616 |
> |  SpeechFlow w/o fine-tuning  |   960 | 2.0  | 0.569 | 0.598 | 2.1 | 0.530 | 0.557 |
> | SpeechFlow w/o fine-tuning, 12-layer Base-size Transformer | 960 | 2.2 | 0.582 | 0.617 | 2.0 | 0.536 | 0.571 |
> | SpeechFlow | 960 | 2.1 | 0.700 | 0.715 | 2.1 | 0.630 | 0.644 |
>
> Combined with the additional results, we believe our experimental results sufficiently support our claims.
>
> ---
>
>
> > The description of the experiments are severely limited. For example, what datasets were used, and the details on the model architecture and hyperparameteres. I have a major concern on reproductivity.
>
> (also associated to first 3 questions raised by the reviewer)
>
> - In all sections from 4.1 (pre-training), 4.2 (enhancement), 4.3 (separation), to 4.4 (zero-shot TTS), we have model/training/data paragraphs with bolded header. Besides description in text, the dataset for each downstream task is provided in the caption of the corresponding Table 1,2, and 3.
> - Here we additionally discuss the data used:
>     - For the 60k hours of unlabeled English speech, we use audiobooks in the public domain. A setup with the same amount of data can be found in [N; cited Wang et al. 2023 in the paper] where the data they used is publicly available.
>     - The 960/360 hours of speech for TTS/separation are also from the same domain, A setup with the same amount of data can be found in [O; cited Ravanelli et al. 2021 in the paper] where the data they used is publicly available.
> - Model architecture and hyper-parameters (learning rate, # of GPUs, etc.) are already provided in Section 4.1, 4.2, 4.3, 4.4 for each task respectively. We understand that this might be hard to track as they are spreaded out in different sections, we have added a summarization in Table 5 in the appendix to make it clear at a glance.
> - In addition to Section 4.6 which already provided analysis on hyper-parameter selection, we have added the study on the model’s robustness w.r.t. masking configuration in Section A.4.5.
> - Finally, we note that the model architecture (24-layer Transformer) and loss function (flow matching) of SpeechFlow is already publicly available [L].
>
> To conclude, we believe we have revealed as many details in the paper as we can to ensure SpeechFlow can be reproduced.
>
> ---

---

> ### Author Response · Authors · 2023-11-17
> **Response to the terminology; Conclusion; Reference**
>
> > Sec 4.4. -- the term "zero-shot TTS" is improper because the model is trained with fully supervised data. the reference is improper either as there are earlier and more established works not cited.
>
> As mentioned in the title and the first paragraph of Section 4.4, **the term “zero-shot TTS” is short for zero-shot speaker adaptation text-to-speech synthesis, not unsupervised/semi-supervised TTS**. Zero-shot TTSs are still trained in a supervised manner. We followed the prior works [D,M,N] to use the term to refer to synthesizing speech with a voice that is unseen during training.
>
> We are sorry if we are missing more early works in this direction, please also kindly provide additional references that should be cited.
>
> ---
>
> **In conclusion, we hope our response has addressed the concerns/questions raised by the reviewer. We sincerely ask the reviewer to re-evaluate this work, especially in terms of novelty. Please kindly provide references if there are still concerns regarding novelty.**
>
> ---
> ### Reference
>
> - [A] Speech Enhancement and Dereverberation with Diffusion-based Generative Models, Ritchard et al.
> - [B] MetricGAN+: An Improved Version of MetricGAN for Speech Enhancement, Fu et al.
> - [C] Attention is All You Need in Speech Separation, Subakan et al.
> - [D] Voicebox: Text-Guided Multilingual Universal Speech Generation at Scale, Le et al.
> - [E] https://voicebox.metademolab.com/zs_tts.html
> - [F] wav2vec 2.0: A Framework for Self-Supervised Learning of Speech Representations, Baevski et al.
> - [G] HuBERT: Self-Supervised Speech Representation Learning by Masked Prediction of Hidden Units, Hsu et al.
> - [H] Emerging Properties in Self-Supervised Vision Transformers, Caron et al.
> - [I] Bootstrap Your Own Latent: A New Approach to Self-Supervised Learning, Grill et al.
> - [J] Exploring the Limits of Transfer Learning with a Unified Text-to-Text Transformer, Raffel et al.
> - [K] Language Models are Unsupervised Multitask Learners, Radford et al.
> - [L] https://github.com/lucidrains/voicebox-pytorch
> - [M] YourTTS: Towards Zero-Shot Multi-Speaker TTS and Zero-Shot Voice Conversion for everyone, Casanova et al.
> - [N] Neural Codec Language Models are Zero-Shot Text to Speech Synthesizers, Wang et al.
> - [O] SpeechBrain: A general-purpose speech toolkit, Ravanelli et al.

---

> ### Comment · Reviewer_dpH8 · 2023-11-23
>
> Thanks for the authors' responses and the additional evaluation and audio samples.
>
>  - Re: subjective evaluation -- thanks for providing these results. I believe that these subjective evaluation results are so important that they should appear in the main content instead of the appendix.
>
>  - Re: audio samples -- thanks for providing the audio samples. I've listened to them and have a question: Compared to one of the baselines SGMSE+, all the samples from SpeechFlow in the speech enhancement group consistently have more artifacts. This seems not aligned with the quantitive evaluation results. Any explanation?
>
>  - To clarify authors' question regarding novelty -- I believe that the problem this paper studies on is highly valuable and relevant, while its novelty and contribution is limited, for the reasons that I stated in the initial review.
>
>  - To clarify authors' question on "speech synthesis" -- speech synthesis is a broader concept than text-to-speech synthesis.
>
>  - I still have concerns regarding the disconnection between the main claim of "foundation model" and the experimental results, especially when finetuning was done with large datasets (360 / 960 hours speech), and the benefits in the experimental results are not very clear (the subjective MOS results in TTS is lower than one of the baselines VoiceBox; the audio samples on speech enhancement sound worse than one of the baselines SGMSE+).
>
> - Re: description of the experiments -- Could you make it clear if the experiments were done with public datasets, and if they are, the name of each specific datasets? Particularly, is the "960 hours" dataset LibriTTS? Similar for other datasets. Please use the name of the public datasets whenever possible.
>
> - While I recognize that the term "zero-shot TTS" is used in a few literatures indeed, it's still not a proper term because it literally means a different task (think about "zero-shot ASR"). For more established prior work: e.g. https://arxiv.org/abs/1806.04558.

---

> > ### Author Response · Authors · 2023-11-23
> > **response to reviewer's clarification and follow up questions (2/2)**
> >
> > > Re: audio samples -- thanks for providing the audio samples. I've listened to them and have a question: Compared to one of the baselines SGMSE+, all the samples from SpeechFlow in the speech enhancement group consistently have more artifacts. This seems not aligned with the quantitive evaluation results. Any explanation?
> >
> > For demo purposes, we use neural vocoder for all the samples and the distortion is mainly coming from the vocoder. This is because the vocoders are not trained on imperfectly-denoised spectrograms and also not trained on Voicebank-Demand data.
> >
> > We have updated the demo page to include samples derived from pseudo-inverse Mel filter bank + phase from noisy speech + iSTFT, where the artifacts are gone. Note that the output Mel spectrogram of our model remains the same, these samples only differ in data post-processing.
> >
> > In addition, we also note that SGMSE+ is indeed a strong baseline that can sometimes sound better than our method, such as the last example in our demo. In this case, signal-processing-based post-processing is even worse than Hifi-GAN, likely due to the poor phase estimation. This showcases how Mel-spectrogram-to-waveform transformation could play an important role in the current framework. However, since it is a separated problem that is beyond the scope and interest of this paper, improving it is left as future work.
> >
> > ---
> >
> > > While I recognize that the term "zero-shot TTS" is used in a few literatures indeed, it's still not a proper term because it literally means a different task (think about "zero-shot ASR"). For more established prior work: e.g. https://arxiv.org/abs/1806.04558.
> >
> > We thank the reviewer for detailing the feedback on terminology. We have replaced all “Zero-shot TTS” with “Zero-shot speaker adaptation TTS” to describe the task more precisely, following the reference provided by the reviewer. We have also added the provided reference to the corresponding text. We hope these updates addressed your concern regarding the terminology adopted from the prior works.
> >
> > ---
> >
> > > Re: subjective evaluation -- thanks for providing these results. I believe that these subjective evaluation results are so important that they should appear in the main content instead of the appendix.
> >
> > We thank the reviewer for the suggestion on improving the paper, **we have included subjective evaluation in the main paper (Section 4.4 and Table 3)** as requested.
> >
> >
> > ---
> >
> >
> >
> > Finally, we thank the reviewer again for the extra effort. We have revised the submission to incorporate the helpful feedback as much as possible within the last 12 hours of discussion period. We hope the concerns have been addressed and we sincerely ask the reviewer to re-evaluate our contribution.

---

> ### Author Response · Authors · 2023-11-23
> **response to reviewer's clarification and follow up questions  (1/2)**
>
> We would like to first thank the reviewer for the clarifications and additional feedback/questions.
>
> ---
>
> > To clarify authors' question regarding novelty -- I believe that the problem this paper studies on is highly valuable and relevant, while its novelty and contribution is limited, for the reasons that I stated in the initial review.
>
> As stated clearly in the paper and throughout the rebuttal, **this is the first work to pre-train a general purpose generative model for speech**.
>
> The initial review referenced in the clarification was “The novelty is limited. It's basically applying flow matching to the speech synthesis problem.”
>
> We understand that the reviewer might not value our solution [1], but **the problem itself, proposed by and studied in this paper, is a novel contribution**. To backup our claim, here we quote reviews from all other reviewers:
> - Reviewer C8mS: *“**Novel idea on making a general-purpose speech generation model** that can perform the following tasks outperforming the current SOTA approaches…”*
> - Reviewer d1Li: *“**The new contribution of this manuscript, the use of generative flow in a self-supervised pre-training stage, is an elegant idea** that forms a strong theoretical paradigm, and that is supported by strong experimental results compared to challenging baselines.”*
> - Reviewer STf2: *“The paper explores **a novel direction to pre-train a general-purpose generative model** with unlabeled speech using flow-based models.”*
>
> In short, this paper introduced a “highly valuable and relevant” (quoting reviewer dpH8’s own word) problem that is novel by itself. And we confirmed that a good solution to this problem can make good contribution to speech generation tasks.
>
>
> [1] Flow matching is the objective we used to train the model, just like how common objectives are used for different purposes (e.g., diffusion for denoising, L1/L2 loss for TTS, etc.). Although using it for separation, denoising, and unsupervised learning with speech is indeed novel, **it is not the main reason why this work is novel**. The main contribution of this paper is exploring pre-trained model for generation tasks.
>
> ---
>
> > Re: description of the experiments -- Could you make it clear if the experiments were done with public datasets, and if they are, the name of each specific datasets? Particularly, is the "960 hours" dataset LibriTTS? Similar for other datasets. Please use the name of the public datasets whenever possible.
>
> For training data  of pre-training/separation/zero-shot speaker adaptation TTS, we use curated datasets that resemble the domain of the test data with different amounts of data as provided. For separation/TTS, the baselines are trained on the same curated datasets, and the comparison is on benchmark datasets (as cited in each table). This setup ensures the evaluation/comparison is fair and convincing while using curated datasets for training.
>
> ---
>
> > I still have concerns regarding the disconnection between the main claim of "foundation model" and the experimental results, especially when finetuning was done with large datasets (360 / 960 hours speech), and the benefits in the experimental results are not very clear (the subjective MOS results in TTS is lower than one of the baselines VoiceBox;
>
> The purpose of the TTS experiment is to show *how much we can reduce the need of labeled data?* and more importantly, *what is the performance cost?*
>
> Voicebox is the fully-supervised strong baseline that requires all the training data of SpeechFlow to be labeled. That’s 60000 hours (6.8 years in total) of speech all labeled. It is expected to be better. But fine-tuning SpeechFlow using only 960 hours provides a good TTS with a very small gap.
>
> We also showed how poorly Voicebox would perform if we restrict the amount of labeled data to be just 960 hours (denoted SpeechFlow w/o pre-training in Table 3).  Between the pre-trained SpeechFlow and randomly initialized Voicebox, there is a significant gap.
>
> The small gap (between data-hungry strong baseline and low-resource fine-tuning) and the big gap (between random initialized v.s. pre-trained model)  are strong evidence that demonstrates how a pre-trained generative speech model is a good foundation model. We show 6250% labeled data can be reduced with a small cost on WER/MOS.

---

### Meta-Review · Area_Chair_1Mhm · 2023-12-04

**Metareview:**

This paper introduces a method of pre-training a flow-based speech synthesizer using 60kh of untranscribed speech, which is then fine-tuned for downstream tasks such as speech enhancement, speech separation, and zero-shot Text-to-Speech (TTS). In this self-supervised paradigm, the challenge of adding labels during the fine-tuning process is addressed by using masked spectrograms as pseudo-labels during pre-training, which are then replaced with actual labels (noisy speech, mixed speech, or phone sequences) during fine-tuning. Three out of four reviewers consistently rated the score above the acceptance threshold. The authors addressed the concerns raised by the reviewer who gave the low score in their rebuttal, but there was no response from that reviewer.

**Justification For Why Not Higher Score:**

The paper’s contributions do not appear to be significant enough, and the methods proposed lack sufficient novelty for a higher score.

**Justification For Why Not Lower Score:**

The authors addressed the concerns raised by the reviewer who gave the low score in their rebuttal, but there was no response from that reviewer.

---

### Decision · Program_Chairs · 2024-01-16

Accept (poster)